

# Overexpression of Myo1e promotes albumin endocytosis by mouse glomerular podocytes mediated by Dynamin

Huijun Shen, Yu Bao, Chunyue Feng, Haidong Fu and Jianhua Mao

Department of Nephrology, The Children's Hospital of Zhejiang University School of Medicine, Hangzhou, Zhejiang, China

Corresponding author
Huijun Shen, 6197005@zju.edu.cn

## ABSTRACT

**Background**. As a fundamental process internalizing molecules from the plasma membrane, endocytosis plays a crucial role in podocyte biology. Our previous study has identified that overexpression of Myo1e may enhance podocyte endocytosis. However, its potential mechanism has been not well understand. Thus, we aimed to analyze whether albumin endocytosis by mouse glomerular podocytes is dependent on Myo1e expression. Also, we aimed to elucidate whether the underlying mechanism is mediated by Dynamin.

**Methods**. Firstly, mouse podocyte cells (MPC5) were treated with different concentrations of FITC-bovine serum albumin (BSA). The fluorescence intensity and cell viability were detected by flow cytometry and MTT assays, respectively. Afterwards, the optimal concentration of FITC-BSA was determined. Secondly, MPC5 cells were treated with Myo1e overexpression or knockdown. Cell morphology was observed under microscope. Immunofluorescence assay was used to determine the expression of F-actin. The protein expression of nephrin and podocin was detected by western blot. Flow cytometry was used to detect MPC5 cell apoptosis with annexin V. Finally, MPC5 cells were treated with Myo1e overexpression and/or Dynasore (a GTPase inhibitor of Dynamin). The fluorescence intensity was detected using flow cytometry assay.

**Results**. MPC5 endocytosis BSA was elevated with a concentration-dependent manner. MTT results showed that MPC5 cell viability was inhibited with a concentration-dependent manner. Myo1e overexpression promoted podocyte endocytic FITC-BSA, which was contrary to its knockdown. Under microscope, after inhibition of Myo1e, podocyte foot process fusion was observed. Myo1e overexpression promoted the expression of cytoskeleton F-actin and podocyte-specific molecules (nephrin and podocin) in podocyte endocytic FITC-BSA. Furthermore, we found that Myo1e promoted the apoptosis of podocytes. Dynasore attenuated the increase in endocytosis of FITC-BSA induced by Myo1e overexpression, suggesting that podocytes might mediate albumin endocytosis via Myo1e-Dynamin-Albumin.

**Conclusion**. Our findings revealed that overexpression of Myo1e promotes albumin endocytosis in mouse glomerular podocyte endocytic albumin mediated by Dynamin.

## INTRODUCTION

Albuminuria is a hallmark of nephropathy, usually caused by a deterioration in the integrity of the glomerular filtration barrier (*Schiessl et al., 2016*). The glomerular filtration barrier consists of a porous endothelium, basement membrane and podocytes. Podocytes are highly specialized and terminally differentiated visceral epithelial cells (*Han et al., 2019*). As an important component of the glomerular basement membrane, podocytes play an important role in maintaining the integrity of the glomerular filtration barrier (*Brosius & Coward, 2014*). *Hartleben et al. (2010)* reported that endoplasmic reticulum stress, excessive ubiquitination, proteinuria and glomerular lesions were observed in aging mice after podocyte-specific knockdown of autophagy-related 5 (Atg5). Furthermore, down-regulation of Cullin-5 expression in UPS as a cytoskeletal protein of E3 ligase in UPS can cause edema, proteinuria and glomerular structural abnormalities in zebrafish, and endoplasmic reticulum stress in podocytes (*Mao et al., 2013*). These finding reveal that it is necessary to explore the mechanisms of abnormally expressed genes in the podocyte cells in the development of albuminuria (*Tryggvason, Patrakka & Wartiovaara, 2006*).

Myosins constitute a large multigene family of actin-based molecular motors in eukaryotes (*Guhathakurta, Prochniewicz & Thomas, 2018*; *Heissler & Sellers, 2016*). According to the amino acid sequence of ATP hydrolysis region, myosins can be divided into 24 categories. Among them, Class I myosin consists of Myo1a~Myo1 h (*Dumont et al., 2002*). Myo1e contains a proline-rich TH2 and a Src-homology 3 (SH3) domain in addition to the TH1 domain (*Cheng, Grassart & Drubin, 2012*). Recent studies have shown that Myo1e is involved in the development of proteinuria. *Krendel et al. (2009)* reported that Myo1e is expressed in glomerular epithelial cells of mice. In mice with Myo1e knockdown, disappearance of podocytes and thickening of the basement membrane are found, eventually leading to proteinuria (*Chase et al., 2012*). Moreover, mutated Myo1e is detected in patients with proteinuria appear (*Mele et al., 2011*; *Sanna-Cherchi et al., 2011*). These studies indicate that Myo1e in podocytes could be involved in the development of proteinuria, which requires further exploration.

Endocytosis is a process of transporting extracellular substances into cells through the deformation movement of the plasma membrane (*Doherty & McMahon, 2009*). Studies have been found that albumin can directly induce podocyte and glomerular injury (*Agrawal & Smoyer, 2017*). As an example, Notch signaling activation promotes the effect of Dynamin-dependent podocyte endocytosis of nephrin, leading to proteinuria (*Waters et al., 2012*). Both in vitro and in vivo studies have found that, podocytes possess the function of endocytic albumin with time- and concentration-dependent manners (*He et al., 2011*). After albumin aggregation in podocytes, knockdown of CD2AP and activation of TRPC6 channels could induce endoplasmic reticulum stress via mediating $Ca^{2+}$ influx, thereby promoting podocyte apoptosis (*He et al., 2011*). These studies demonstrate that the endocytic BSA in podocytes is associated with the formation of proteinuria, but the specific mechanisms are poorly understood. Based on above studies, we hypothesized that overexpression of Myo1e could promote albumin endocytosis by mouse glomerular podocytes mediated by Dynamin.

## MATERIALS AND METHODS

### Cell culture

Murine kidney podocyte cell line MPC5 (Mount Sinai School of Medicine, New York, NY) were cultured in RPMI-1640 medium (SH30809.01B, Hyclone, China) plus 10% fetal bovine serum (SH30084.03, Hyclone, China) and 10 U/ml mouse recombinant interferon-γ at 33 °C with 5% $CO_2$. To induce differentiation, MPC3 cells were cultured for 14 days at 37 °C without interferon-γ.

### Endocytosis

MPC5 cells were seeded in 6-well plates ($2.5^*10^5$/well) in three duplicates. After 24 h, the cells were treated with different concentrations (0 ug/ml, 100 ug/ml, 250 ug/ml, 500 ug/ml, 1 mg/ml) of FITC-BSA (SF063, Solarbio, China) for 4 h in the dark. Flow cytometry was used to evaluate the endocytosis.

### MTT assay

The MPC5 cells were seeded in $7^*10^3$/well. 200ul of FITC-BSA (0 ug/ml, 100 ug/ml, 250 ug/ml, 500 ug/ml, 1 mg/ml) and RPMI-1640 medium suspension was added to each well, and cultured at 37 °C for 4 h. After medium change, the cells were cultured for another 24 h. After that, the cells were treated with 20 ul MTT solution (5 mg/ml; M5655-1G, Sigma, USA) for 4 h at 37 °C. 150 ul of DMSO was then added to each well for 10 min. Absorbance was detected at 490 nm.

### Plasmid transfection

The Myole- or Dynamin- plasmid transfection was performed as our previous study (*Jin et al., 2014*). The shRNA targeting Myole (shMyole) and full length of Myole overexpression were synthesized in Invitrogen (Carlsbad, CA, USA). Furthermore, siRNAs targeting Dynamin (siDynamin) and full length of Dynamin overexpression were also synthesized. Then, the target sequences were cloned into pLenti6.3_MCS_IRES2-EGFP lentiviral vector. The lentiviral vectors with the target sequence of shMyole or siDynamin were transfected into HEK293FT cells. MPC5 cells were transfected with negative control shRNA (shNC), shMyole, empty pcDNA3.1 plasmid (empty vector), Myole-plasmid (Myole overexpression), negative control siRNA, siDynamin and Dynamin overexpression using Lipofectamine 2000 (Invitrogen), respectively. The cell morphology was observed under an optical microscope.

### Real-time quantitative PCR (RT-qPCR) analysis

Total RNA was extracted from MPC5 cells using TaKaRa MiniBEST Universal RNA Extraction Kit (TaKaRa, Beijing, China). The cDNA was synthesized with M-MLV (Vazyme, Jiang Su, China). RT-qPCR was carried out using SYBR Green I Master (Roche, Beijing, China). The primers were as follows: Mus Myo1e, 5′-ACAGTGCGCAACAACAACTC-3′ (forwards), 5′-TGATGCCAAGGCTTTGCTTC-3′ (reverse); Mus Dynamin, 5′-TTTGCCAATGCTCAGCAGAG-3′ (forwards), 5′-TGCTTGCTTGACATGAAGCC-3′ (reverse); Mus GAPDH, 5′-TGATGCCAAGGCTTTGCTTC-3′ (forwards), 5′-GATGGCAACAATCTCCACTTTG-3′ (reverse). GAPDH was used as an internal control. The results were calculated using the $2^{-\Delta\Delta}$ method.

## Immunofluorescence assay

The cells were seeded into a 24-well plate ($1*10^5$/well) at 37 °C. After 24 h of transfection with empty vector, Myole-plasmid, shNC and shMyole, the medium was removed from the 24-well plate. After washing twice in PBS, the expression of F-actin was stained by phalloidin and detected in MPC5 cells under a fluorescence microscopy (Olympus).

## Western blot analysis

The MPC5 cells transfected with empty vector, Myole-plasmid, shNC and shMyole were lysed with RIPA buffer on ice for 10–20 min, and centrifuged at 12,000 rpm at 4 °C for 3–5 min. Transfer the supernatant to a new 1.5 ml EP tube. The protein in the supernatant was collected for western blot analysis. After separating on SDS-PAGE, the protein was transferred onto PVDF membrane. After that, the membrane was blocked using 5% milk for 1 h at room temperature. The primary antibody was diluted with 1% BSA/PBST, and the membrane was incubated overnight at 4 °C in a refrigerator. Afterwards, the PVDF membrane was incubated with a horseradish peroxidase-labeled secondary antibody diluted in 5% milk/PBST for 1 h at room temperature. ECL was used to visualize the protein blots. The primary antibodies included goat anti-nephrin (1:100, Santa Cruz Biotechnology) and anti-podocin (1:100, Santa Cruz Biotechnology). The relative expression levels were corrected for GAPDH expression.

## Flow cytometry for apoptosis

The MPC3 cells were plated in 6-well plates ($2.5*10^5$ cells/well) at 37 °C and 5% $CO_2$. After 24 h, Myole-plasmid and empty vector were transfected into MPC3 cells, respectively. After transfection for 24 h, the cells was treated with Dynasore (160 uM) for 24 h. After that, the cells were washed twice with PBS, and then added 500 ul of serum-free RPMI-1640 and different concentrations of 500 ug/ml FITC-BSA for 4 h. Finally, the cells were collected for flow cytometry apoptosis assay. According to the manufactures' protocols, apoptotic MPC5 cells were stained with Annexin V-FITC/Propidium Iodide (PI) (Beckman Coulter Trading Co., Ltd., China, Shanghai), and then analyzed on CytoFLEX S Flow Cytometer (BD Biosciences).

## Statistical analysis

Graphpad Prism 7.0 (San Diego, CA, USA) was used for statistical analyses. All experiments were independently repeated at least three times. The data were presented as mean $\pm$ standard deviation (SD). Comparisons between two groups were analyzed with paired student's t test. For pairwise multiple comparisons, one-way analysis of variance (ANOVA) was performed, followed by Tukey's multiple comparison test. $P$-value<0.05 was considered statistically significant.

# RESULTS

## The optimal concentration of BSA when treating MPC5 cells

We first determined the optimal concentration of endocytic BSA in podocytes. Flow cytometry was used to detect changes in fluorescence intensity after treatment with

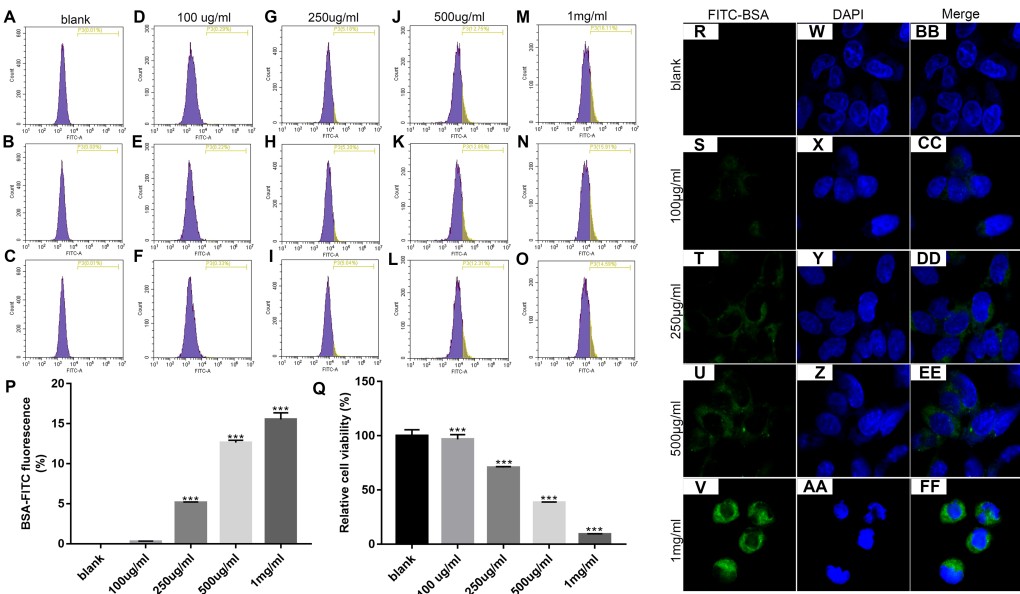

**Figure 1** **Identification of the optimal concentration of BSA when treating MPC5 cells.** (A–P) Flow cytometry assay results showing the endocytosis effects of MPC5 cells treated with different concentrations (0 ug/ml, 100 ug/ml, 250 ug/ml, 500 ug/ml, 1 mg/ml) of FITC-BSA. (Q) The cell viability of MPC5 cells treated with different concentrations (0 ug/ml, 100 ug/ml, 250 ug/ml, 500 ug/ml, 1 mg/ml) of FITC-BSA was evaluated using MTT assay. (R–FF) Immunofluorescence showing the endocytosis effects of MPC5 cells treated with different concentrations of FITC-BSA. Scale bar, 50 μm. Compared to blank, ***$p < 0.001$.

different concentrations of FITC-BSA for MPC-5 podocytes. The results showed that the higher the concentration of BSA, the higher the content of BSA phagocytized by MPC-5. When the concentration of BSA was 100 ug/ml, the phagocytosis content of MPC5 was the lowest; when the concentration of BSA was 1 mg/ml ($p < 0.001$), the phagocytosis content of MPC5 was the highest (Figs. 1A and 1B). Next, we examined the effect of BSA on the proliferation of MPC-5 podocytes. The proliferation of cells was detected by MTT assay after treatment with different concentrations of BSA for MPC-5. The results showed that BSA-FITC (250 ug/ml, 500 ug/ml and 1 mg/ml) significantly inhibited MPC5 cell apoptosis ($p < 0.001$; Fig. 1C). Immunofluorescence results showed that the content of BSA phagocytized by MPC-5 was increasing depending on the concentration of BSA, which was consistent with flow cytometry results (Fig. 1D). Combined with above results, 500 ug/ml BSA was selected for subsequent experiments.

## Overexpression of Myo1e enhances MPC5 glomerular podocyte endocytosis BSA

We further observed the effects of Myo1e on endocytosis BSA. RT-qPCR results showed that Myo1e was successfully overexpressed (Fig. 2A; $p < 0.001$) and inhibited (Fig. 2B; $p < 0.01$). Flow cytometry results showed that MPC5 endocytosis BSA increased significantly after overexpression of Myo1e compared to empty vector group ($p < 0.001$), however, MPC5 endocytosis BSA was significantly decreased after Myo1e knockdown compared to shNC

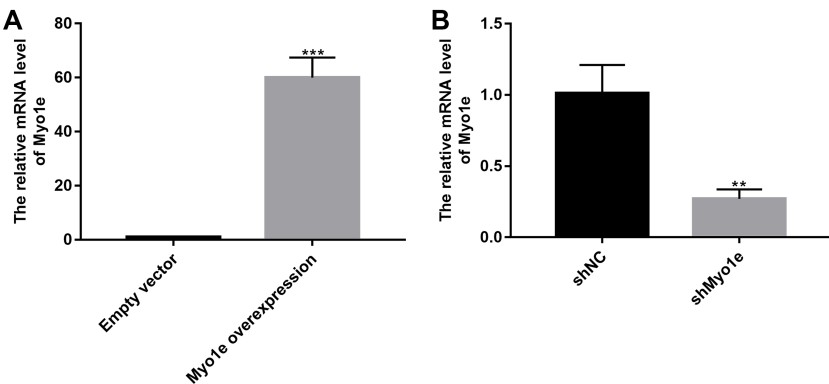

**Figure 2 The effects of Myo1e overexpression or knockdown.** RT-qPCR results showed that Myo1e was successfully overexpressed (A) or inhibited (B). ** $p < 0.01$; *** $p < 0.0001$.

group ($p < 0.01$; 3A and 3B), which was consistent with immunofluorescence results (Fig. 3C). The above results indicated that Myo1e may promote MPC5 endocytic BSA. In addition, we observed the effect of abnormally expressed Myo1e on the morphology of MPC5 cells. As shown in Fig. 4, compared with empty vector group, when Myo1e was overexpressed, we observed that the MPC5 cell morphology was irregular, the somas became larger, the foot processes were thinner and longer, and some of the podocytes also had a dendritic bifurcation. After the knockdown of Myo1e, the somas were shrunk, the edges were not neat, and the cells were fused.

### Myo1e may promote the expression of F-actin in MPC5 cells

We further analyzed the effect of Myo1e overexpression/knockout on F-actin expression in MPC5 cells. The immunofluorescence results showed that the fluorescence intensity of F-actin increased after overexpression of Myo1e gene compared with empty vector group (Fig. 5). However, the fluorescence intensity of F-actin decreased when Myo1e expression was inhibited compared to shNC group. The experimental results suggested that Myo1e may promote the expression of F-actin in MPC5 cells.

### Myo1e may promote the expression of nephrin and podocin in MPC5 cells

The western blot analysis results showed that the expression of nephrin and podocin increased after Myo1e overexpression compared with empty vector group (Figs. 6A–6C). However, the expression of nephrin and podocin decreased when knockdown Myo1e compared to shNC group (Figs. 6A–6C). Our results indicated that the Myo1e may promote the expression of nephrin and podocin in MPC5 cells.

### Myo1e may promote MPC5 cell apoptosis

The flow cytometry results showed that compared with the empty vector group, after overexpression of Myo1e, the apoptosis of MPC5 cells significantly increased ($p < 0.05$; Figs. 7A and 7B). However, the apoptosis of MPC5 cells significantly decreased when Myo1e

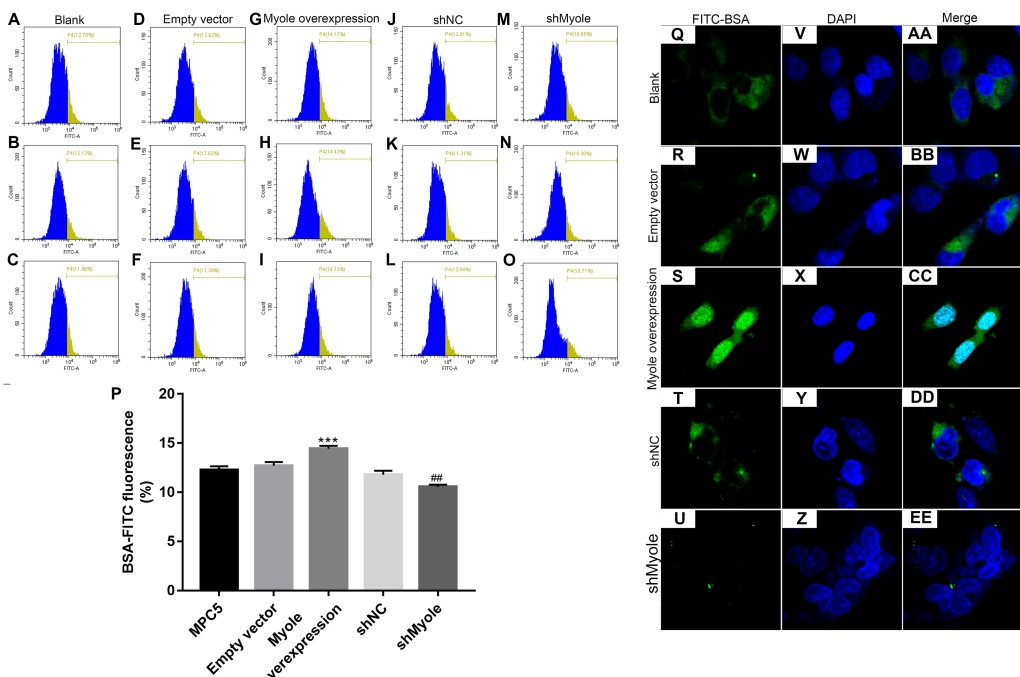

**Figure 3  Overexpression of Myo1e enhances MPC5 glomerular podocyte endocytosis BSA.** (A–P) Flow cytometry assay results showing the endocytosis effects of MPC5 cells. (Q–EE) Immunofluorescence results showing the endocytosis effects of MPC5 cells. Scale bar, 50 μm. ***Myo1e overexpression vs. empty vector, $p < 0.001$; ##shMyo1e vs. shNC, $p < 0.01$.

knockdown compared to shNC group ($p < 0.001$; Figs. 7A and 7B). The experimental results indicated that Myo1e may promote apoptosis of MPC5 cells.

## Myo1e might promote MPC5 endocytosis BSA mediated by Dynamin

We further explored how Myo1e enhanced glomerular podocyte endocytosis of BSA. From the experimental results, it was found that MPC5 endocytosis BSA was significantly reduced after MPC5 cells treated with a GTPase inhibitor of Dynamin, Dynasore (Figs. 8A and 8B). In addition, compared with the MPC5 + Myole overexpression group, MPC5 endocytosis BSA was significantly inhibited after MPC5 cells treated with Dynasore (Figs. 8A and 8B). The effect of Dynosore on both Myo1e and Dynamin expression was examined using RT-qPCR. The results showed that Dynosore significantly inhibited the expression of Dynamin (Fig. 8C), however, it did not affect the expression of Myo1e (Fig. 8D). Above results indicated that Dynasore can inhibit MPC5 endocytosis BSA.

As shown in Fig. 9A, RT-qPCR results showed that Dynamin was successfully overexpressed ($p < 0.001$). Furthermore, three siRNAs targeting Dynamin was synthesized. RT-qPCR results showed that MPC5 cells treated with Dynamin-siRNA2 had the lowest expression levels of Dynamin ($p < 0.001$; Fig. 9B). Thus, Dynamin-siRNA2 was used for further experiments. Immunofluorescence results showed that Dynamin knockdown inhibited MPC5 endocytosis BSA, while its overexpression promoted MPC5 endocytosis BSA (Fig. 10). Furthermore, Dynamin knockdown decreased the effect of

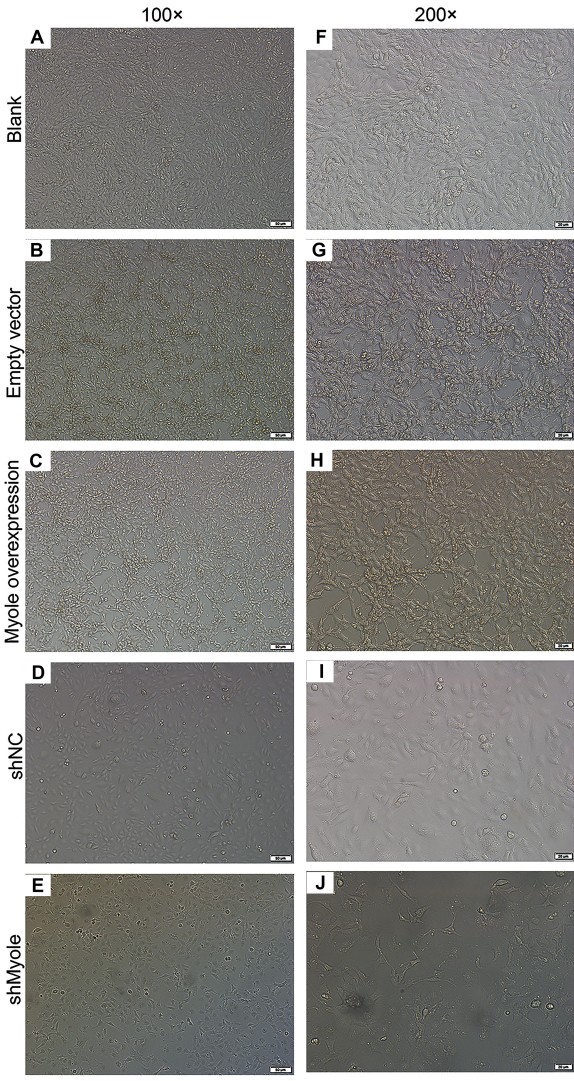

**Figure 4  Morphology changes of MPC5 cells when overexpression or knockdown of Myo1e ((A–E) 100×, (F–J) 200×).**

MPC5 endocytosis BSA induced by Myo1e overexpression (Fig. 10). We also found that Dynamin overexpression ameliorated the inhibitory effect of MPC5 endocytosis BSA induced by Myo1e knockdown (Fig. 10). These results indicated that podocytes mediate albumin endocytosis through Myo1e-Dynamin-albumin.

## DISCUSSION

Proteinuria, mainly characterized by albuminuria, is not only a marker but also a known risk factor for progressive glomerular disease. Tubulointerstitial damage has been a field of widespread interest in this animal model. In addition, *in vitro* studies have revealed the role of serum albumin and its binding factors as mediators of proximal tubule cell damage, however, its molecular role in podocytes is not well understood. The response of

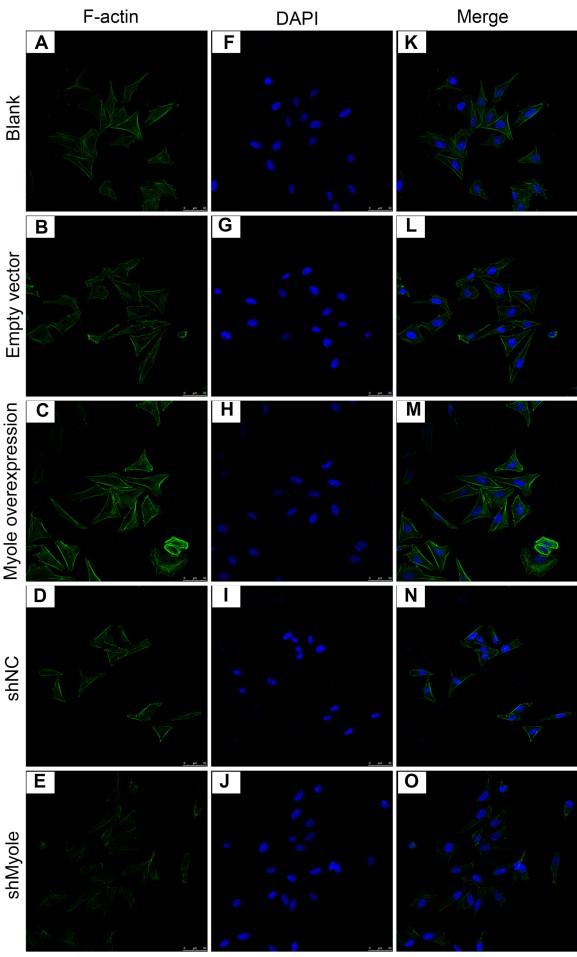

**Figure 5** Immunofluorescence detection of F-actin expression in MPC5 cells treated with Myo1e over-expression or knockdown (A–O). Scale bar, 50 μm.

podocytes to serum albumin includes albumin endocytosis and apoptosis. Myo1e plays an important role in renal function. Previous research has reported that the podocyte-specific knockout myo1e was performed using Cre-mediated recombination controlled by the podocin promoter (*Chase et al., 2012*). Loss of Myo1e in podocytes results in proteinuria, disappearance of the podocyte process and disintegration of the glomerular basement membrane. Podocytes can endocytose proteins, including albumin, immunoglobulins and transferrin, in a receptor-mediated manner. In our previous studies, we analyzed endocytic FITC-transferrin by podocyte analysis by quantitative analysis and fluorescence microscopy. After co-culture of podocytes with FITC-transferrin, the number of cells with FITC-positive vesicles in somatic cells treated with Myo1e knockdown was significantly decreased. However, FITC-transferrin was observed in abundant large vesicles in podocytes, especially in podocytes overexpressing Myo1e. Our previous study indicated that inhibition of Myo1e expression may reduce the efficiency of endocytic FITC-transferrin in podocytes. Our previous study has identified that Myo1e was expressed in the mouse podocytes

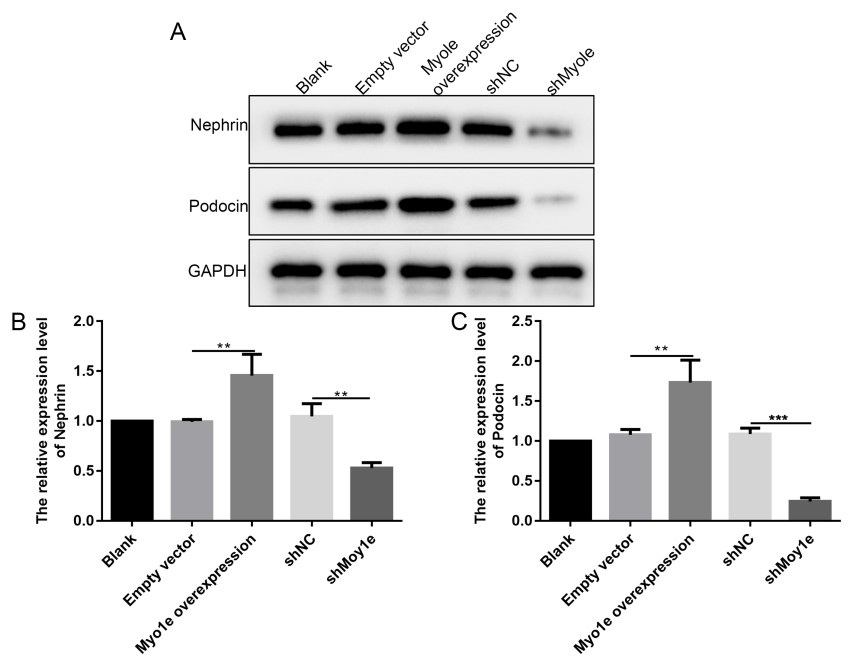

**Figure 6** **Western blot analyses showing the expression levels of nephrin and podocin in MPC5 cells treated with Myo1e overexpression/knockdown.** (A) Representative images of protein blots. (B) The expression levels of nephrin. (C) The expression levels of podocin. $**p < 0.01$; $***p < 0.001$.

of glomeruli, furthermore, overexpression of Myo1e may promote cellular endocytosis, migration, and adhesion (*Jin et al., 2014*). However, the specific mechanisms remain unclear. BSA is a main porotein component of proteinuria, therefore, in our current study, we observed the podocyte endocytosis of FITC-BSA by fluorescence microscopy in a concentration-dependent manner. The MTT assay showed that FITC-BSA inhibited podocyte proliferation in a concentration-dependent manner.

In this study, we found that overexpression/knockdown of Myo1e can cause changes in the function and morphology of endocytic albumin in podocytes. Our results showed that overexpression of Myo1e promoted the ability of podocytes to endocytosis and while knockdown of Myo1e inhibited the ability of podocytes to endocytosis. Renal biopsy in patients with proteinuria usually manifests as the disappearance of podocyte foot processes. We found that MPC5 cells treated with knockdown of Myo1e appeared foot process fusion, which was contrary to MPC5 cells treated with overexpression of Myo1e. Myo1e may ameliorate podocyte foot process fusion of patients with proteinuria (*Perysinaki et al., 2011*).

F-actin cytoskeletal disruption is a typical characteristic of podocyte injury. F-actin cytoskeleton has been shown to be critical for maintaining the specific morphology of podocyte foot processes (*Allison, 2015*; *Hu et al., 2017*; *Schiffer et al., 2015*). Destruction of the F-actin cytoskeleton in podocytes results in the disappearance of the foot process and is associated with the pathogenesis of proteinuria (*Ni et al., 2018*). In our study, immunofluorescence results showed that Myo1e overexpression promoted the expression

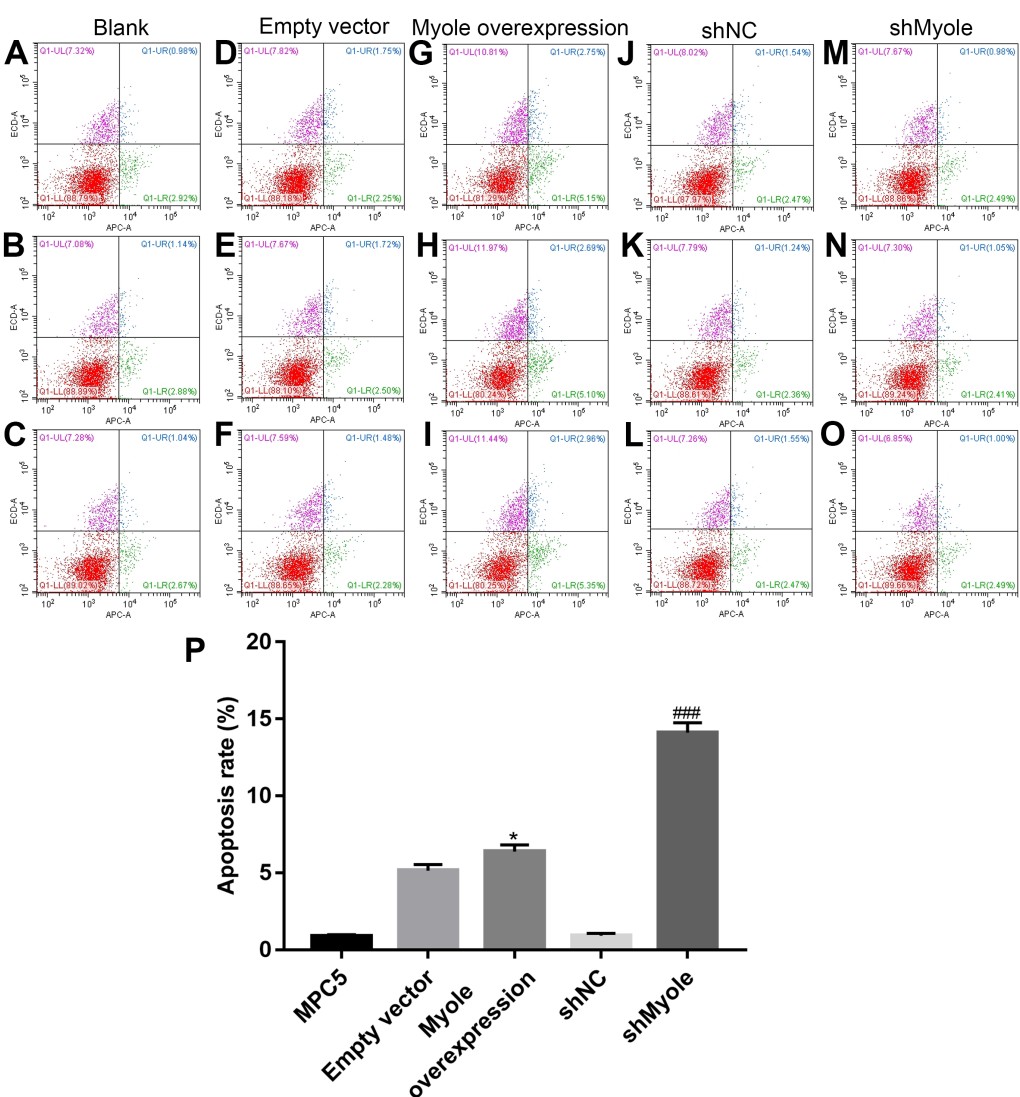

**Figure 7  Myo1e may promote MPC5 cell apoptosis.** (A–P) Flow cytometry showing the apoptosis of MPC5 cells when treated with Myo1e overexpression/knockdown. * Myo1e overexpression vs. empty vector, $p < 0.05$; ### shMyo1e vs. shNC, $p < 0.001$.

of F-actin in MPC5 cells, which was contrary to its knockdown. Thus, Myo1e may play an important role in podocyte endocytosis by regulating the actin cytoskeleton F-actin.

Slit membrane proteins are essential molecular components of the glomerular filtration barrier and are also involved in actin polymerization (*Kim, Kim & Kim, 2016*). As podocyte-specific proteins, nephrin and podocin play important roles in the function of the glomerular filtration barrier (*Ni et al., 2017*). The disappearance of the foot process may be due to diaphragm rupture and podocyte damage. Nephrin is a structural component of a slit membrane formed by adjacent podocytes (*Ruotsalainen et al., 1999*). The absence of nephrin contributes to proteinuria and foot process effacement (*Teng et al., 2016*). Podocin is a key factor in maintaining the steady state of the slit diaphragm. Our results showed

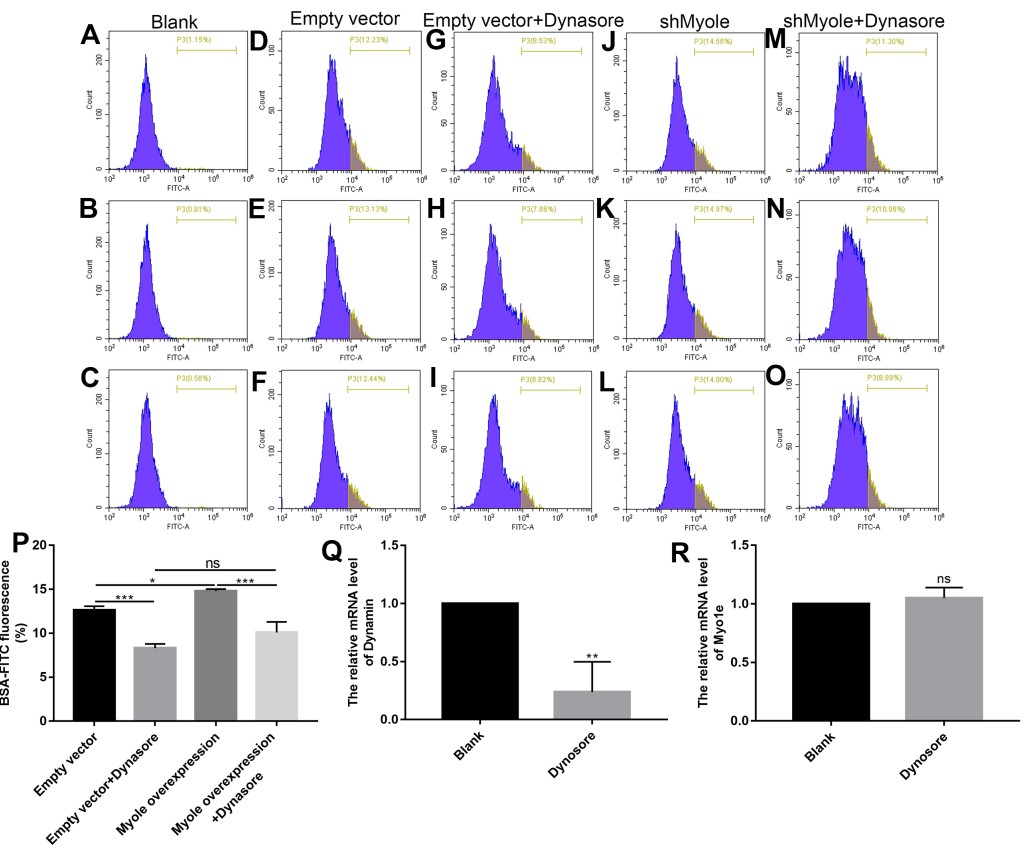

**Figure 8** **Myo1e might promote MPC5 endocytosis BSA mediated by Dynamin.** (A–P) Flow cytometry showing the endocytosis effects of MPC5 cells. (Q, R) RT-qPCR showing the effect of Dynosore on both Myo1e and Dynamin expression. * $p < 0.05$; ** $p < 0.01$; *** $p < 0.001$.

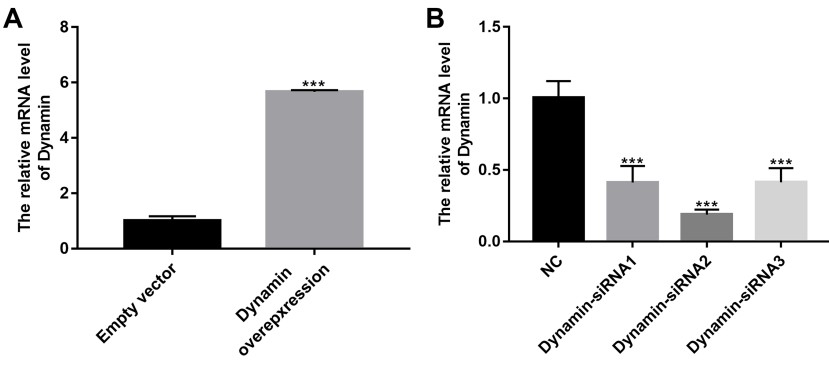

**Figure 9** **The effects of Dynamin overexpression or knockdown.** RT-qPCR results showed that Dynamin was successfully overexpressed (A) or silenced (B). *** $p < 0.0001$.

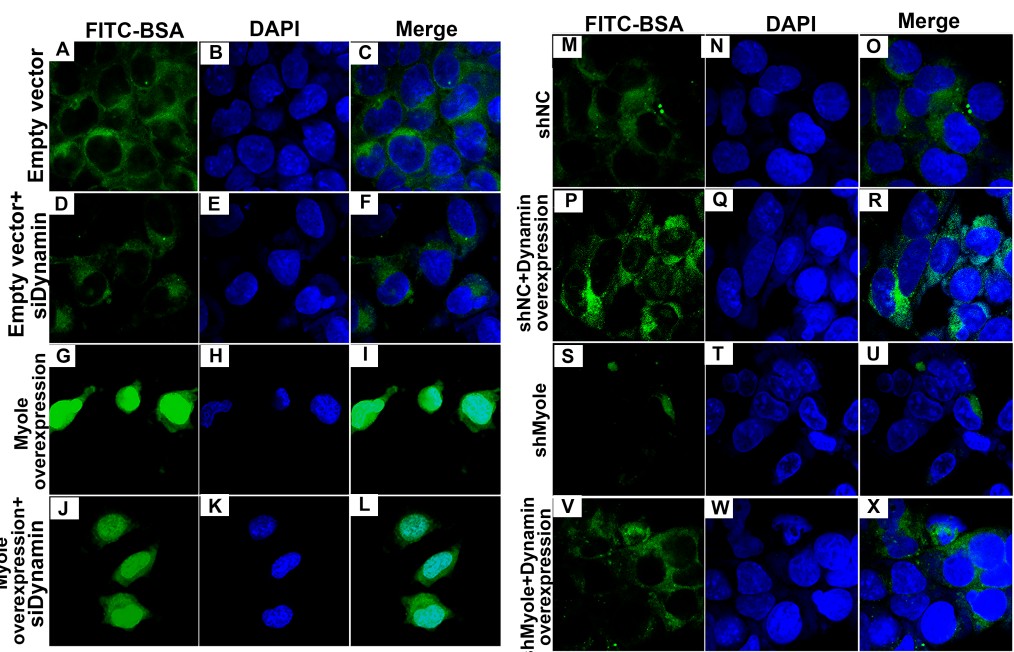

**Figure 10** **Immunofluorescence showing Myo1e might promote MPC5 endocytosis BSA mediated by Dynamin (A–X). Scale bar, 50 μm.**

that overexpression of Myo1e promoted the expression of nephrin and podocin in MPC5 cells. However, the expression of nephrin and podocin decreased when knockdown of Myo1e in MPC5 cells. These results indicated that Myo1e may be involved in maintaining slit membrane by nephrin and podocin. Increasing evidence suggests that the expression of nephrin and podocin could be regulated by many factors. For example, microRNA-29a may promote nephrin acetylation to improve hyperglycemia-induced podocyte dysfunction (*Lin et al., 2014*). Another research found that vitamin D3 ameliorates podocyte injury via targeting nephrin (*Trohatou et al., 2017*).

Podocyte injury has been shown to contribute to the development of proteinuria (*Li, Ma & Liu, 2019*). Our findings showed that overexpression of Myo1e promoted apoptosis of MPC5 cells that were co-cultured with BSA, indicating that overexpression of Myo1e may induce podocyte injury. There are several pathways of albumin endocytosis, such as Dynamin-dependent podocyte endocytosis, FcRn-mediated albumin transcytosis (*Kinugasa et al., 2011*), clathrin-mediated endocytosis (*Soda et al., 2012*), caveolin-mediated endocytosis (*Dobrinskikh et al., 2014*), fluid phase-endocytosis (*Palm et al., 2015*) and dynein-microtubule related vesicle transport (*Tojo et al., 2017*). Dynamin plays a crucial role in maintaining the structure and function of the glomerular filtration barrier. Dynamin regulates the actin cytoskeleton and the turnover of nephrin in podocytes, furthermore, knockdown of dynamin leads to proteinuria (*Khalil et al., 2019*). Dynamin-dependent podocyte endocytosis is one of the pathways of albumin endocytosis. In our study, we found that Dynasore attenuated the increase in endocytosis of albumin induced

by Myo1e overexpression. As a GTPase inhibitor of Dynamin, these results indicated that podocytes might mediate albumin endocytosis by Myo1e-Dynamin-Albumin.

## CONCLUSION

In our study, our results showed that Myo1e promoted podocyte endocytic albumin, however, after inhibition of Myo1e, podocyte foot process fusion was observed. Furthermore, we found that Myo1e promoted apoptosis of podocytes. Myo1e elevated the expression of podocyte-specific molecules (nephrin and podocin) and cytoskeleton F-actin in podocyte endocytic albumin. Dynasore attenuated the increase in endocytosis of albumin induced by Myo1e overexpression, suggesting that podocytes might mediate albumin endocytosis via Myo1e-Dynamin-Albumin.

### Abbreviations

| | |
|---|---|
| **BSA** | bovine serum albumin |

### Funding
This work was funded by the Zhejiang Provincial Natural Science Foundation of China (LH14H050002). The funders had no role in study design, data collection and analysis, decision to publish, or preparation of the manuscript.

### Grant Disclosures
The following grant information was disclosed by the authors:
The Zhejiang Provincial Natural Science Foundation of China: LH14H050002.

### Competing Interests
The authors declare there are no competing interests.

### Author Contributions
- Huijun Shen conceived and designed the experiments, performed the experiments, prepared figures and/or tables, and approved the final draft.
- Yu Bao performed the experiments, prepared figures and/or tables, and approved the final draft.
- Chunyue Feng performed the experiments, authored or reviewed drafts of the paper, and approved the final draft.
- Haidong Fu and Jianhua Mao analyzed the data, authored or reviewed drafts of the paper, and approved the final draft.

### Data Availability
The raw measurements are available in the Supplemental Files.

## Supplemental Information

Supplemental information for this article can be found online at http://dx.doi.org/10.7717/peerj.8599#supplemental-information.

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
