# Peer review of "Overexpression of Myo1e promotes albumin endocytosis by mouse glomerular podocytes mediated by Dynamin"

_PeerJ, doi:10.7717/peerj.8599_

## Round 0.1 · original submission · Major Revisions

As you will see, your submission got two reviews of high quality, one of them being particularly well detailed. I myself would concur with the comment raised by the other reviewer about the risk of getting false results by relying only on fluorescence data to describe the fate of albumin. The problem is that FITC-albumin could it-self have a different behaviour than unlabeled albumin while, as pointed out by the reviewer, FITC could also be released, making the determination of florescence meaningless.

My view and decision at this point is that you should provide satisfactory answers to the remarks raised by both reviewers. If you do so, please, submit a detailed rebuttal where you indicate how (and where in the revised version) you have taken each comment into consideration. If you do not agree with a comment, please, explain why. Your rebuttal will be an essential element for me to take a final decision. Please, be also aware that your revised version will undergo a new round of review by the same or by different reviewers. I cannot, therefore, make any commitment about a final acceptance of your paper, but look forward to see it again whenever you are ready. If you need additional delay, inform the Editorial Office so that we can take note of it.

·

Basic reporting

1. The English language should be improved to ensure that an international audience can clearly understand your text. Examples include the title and research question as posed in the abstract (.. .whether Myo1e albumin endocytosis in mouse glomerular podocyte enocytic albumin and it possible mechanism). Other examples include lines 78, 82, 85, and 93.
Also, I would advise to refrain from using sentences such as ‘and so on’ (line 55)‘and other functions’(line 86)
I would advise using a professional editing service to ensure a thorough improvement, which is needed.

2. The text in the introduction (especially lines 54 to 68) is very difficult to comprehend and contains many references that don’t seem immediately relevant to fill a knowledge gap for your particular research question. It isn’t necessary to provide a comprehensive review of all the functions podocytes have in this regard, as this would become quite extensive. However, some key elements seem to be missing now. Stating that the role of podocytes in proteinuria can be attributed to ‘nephrin and podocin and so on’ does not provide enough information. Also, there is extensive literature on nephrin and podocin and proteinuria (NEJM 2006; 354 Tryggvason et al for example). Placing only self references here does not seem thorough enough.
Please consider rewriting this section of the introduction. Consider including reviews on the role of podocytes in the development of proteinuria, endocytosis in podocytes in proteinuria/maintenance of the GFB and podocyte apoptosis in proteinuric diseases. For example, see Nagata’s Kidney International paper (PMID 27165817)

3. In the introduction (line 49), the author states that damage to podocytes inevitably leads to albuminuria. This is not necessarily true, as for example increased reabsorption by the PTC can take place. Also, not all damage to the GFB is irreversible. An increase in glomerular permeability can occur without leading to proteinuria.

4. In line 240 the author claims that little research has been done on podocyte molecular changes. The paper’s own reference list contradicts this statement. Please revise.

Experimental design

1. Please clearly state the research question in the introduction (and abstract). The current research question formulated in the abstract reads: ‘we aimed to analyze whether Myo1e albumin endocytosis in mouse glomerular podocyte endocytic albumin and its possible mechanism.’
For example: we aimed to analyze whether albumin endocytosis by mouse glomerular podocytes is dependent on Myo1e expression. Also, we aimed to elucidate whether the underlying mechanism is mediated by Dynamin.
2. To allow full comprehension of your data, please include mean values, SD and the p value (or CI) when reporting your results.
3. Validation and quantification of your Myo1e overexpression and shMyo1e experimental models is necessary for proper interpretation of results. Please add these experiments (Myo1e expression levels) to the current paper.
4. Are relative expression levels as shown in Figure 5 corrected for GAPDH expression? Is yes, please state this in your methods.
5. Figure 7B shows that the effect of Dynasore on endocytosis is much greater than the effect of Myo1e overexpression. Please provide statistical comparison between the empty vector+dynasore and Myo1e overexpression+dynasore groups.
6. Validation of the effect of Dynosore on both Myo1e and Dynamin expression should be added to allow for a proper interpretation of results (dynamin expression levels). Please add this to the current paper.
7. Figure 3: please provide images in a higher magnification. The current image is hard to appreciate. Please highlight the morphological changes you want the reader to pay attention to (larger somas, foot process effacement, dendritic bifurcation)

Validity of the findings

1. The relative increase in BSA-FITC fluorescence in Myo1e overexpression and decrease in BSA-FITC fluorescence in Myo1e knockdown is significant, but small. Please elaborate on whether you expect this increase to be clinically relevant. Your results seem to show that Myo1e is involved in albumin endocytosis, but that it does not necessarily play a large or essential role.
2. Additional experiments are required to substantiate the hypothesis that the effect of Myo1e is mediated by dynamin.
Why did you choose to use dynasore and not a direct dynamin knockdown or knockout? As stated by Preta et al. in Cell Commun Signal 2015; 13: 24: Dynasore provides rapid and reversible inhibition of dynamin-dependent endocytosis, which is effective in cells from several species. However, in addition to inhibition of the GTPase of dynamin, dynasore has wider effects on cellular cholesterol, lipid rafts, and actin.
3. Please elaborate as to why the observed increase in apoptosis in podocytes with Myo1e overexpression occurs and how this would lead to amelioration of podocyte injury (line 293). This statement seems contradictory.

Additional comments

In 'overexpressoin of Myo1e promotes albumin endocytosis in mouse glomerular podocyte endocytic albumin mediated by Dynamin’, Shen et al. Show that Myo1e plays a role in albumin endocytosis by podocytes, possibly mediated by Dynamin in a series of in vitro experiments.
The content of this paper shows merit and seems scientifically valid. However, there are some areas where the article could be improved. The major points of concern are stated above. The most serious points of concern are those mentioned under basic reporting point 1, and all points stated under validity of the findings.

Reviewer 2 ·

Basic reporting

.

Experimental design

.

Validity of the findings

.

Additional comments

The authors investigated the role of Myo1e in albumin endocytosis in the mouse glomerular podocyte culture cells.

Comments
1. FITC-albumin is easy to dissociate and free FITC will affect the results (Medical Molecular Morphology 2008; 41 (2) 92-98). Have you check how many % of free FITC exist in each concentration of FITC-albumin solution? In Figure 1, if scales of Y-axis were same in each FITC-albumin concentration, total amount of FITC-albumin endocytosis may be not so increased and may become a plateau in 1mg/ml. To confirm the real endocytosis of FITC-albumin, the fluorescent microscopic pictures of FITC-albumin in MPC5 cell culture should be demonstrated in each concentration.
2. In Figure 2, as the scales of Y-axis are different, it is difficult to see the absolute increase of endocytosis. Thus, increased endocytosis of FITC-albumin should be confirmed by immunofluorescence microscopy of FITC-albumin in Myo1e overexpressed culture cells. Figure 3 is difficult to see the morphological changes without some staining. It is much better to show the immunofluorescence microscopy to show the change in endocytosis vesicles.
3. In Figure 5B, C there are no SD bars, how many experiments were repeated?
4. Dynamin-dependent podocyte endocytosis is one of the pathways of albumin endocytosis. The other possibilities of podocyte albumin transport should be discussed; FcRn-mediated albumin transcytosis (Kidney Int 2011; 80: 1328–1338), clathrin-mediated endocytosis (JCI 2012), caveolin-mediated endocytosis (Am J Physiol Renal Physiol 2014;306:F941-951), fluid phase-endocytosis (2015) and dynein-microtubule related vesicle transport (Med Mol Morphol. 2017; 50(2):86-93), and so on.

---

## Round 0.2 · accepted · Accept

You revised version was considered satisfactory and I guess the paper is now much stronger.

Reviewer 2 ·

Basic reporting

see below

Experimental design

see below

Validity of the findings

see below

Additional comments

The authors responded well to all my comments, so I agree to accept the manuscript.

There is a typo: Line 19, 26 and 30, Myole should be Myo1e.